# Co-Formulation of Amphiphilic Cationic and Anionic Cyclodextrins Forming Nanoparticles for siRNA Delivery in the Treatment of Acute Myeloid Leukaemia

**DOI:** 10.3390/ijms23179791

**Published:** 2022-08-29

**Authors:** Ayse Kont, Monique C. P. Mendonça, Michael F. Cronin, Mary R. Cahill, Caitriona M. O’Driscoll

**Affiliations:** 1Pharmacodelivery Group, School of Pharmacy, University College Cork, T12 YN60 Cork, Ireland; 2Department of Haematology and CancerResearch@UCC, Cork University Hospital, University College Cork, T12 XF62 Cork, Ireland

**Keywords:** modified cyclodextrins, nanomaterials, nucleic acids, non-viral gene delivery, acute myeloid leukaemia (AML)

## Abstract

Non-viral delivery of therapeutic nucleic acids (NA), including siRNA, has potential in the treatment of diseases with high unmet clinical needs such as acute myeloid leukaemia (AML). While cationic biomaterials are frequently used to complex the nucleic acids into nanoparticles, attenuation of charge density is desirable to decrease in vivo toxicity. Here, an anionic amphiphilic CD was synthesised and the structure was confirmed by Fourier-transform infrared spectroscopy (FT-IR), Nuclear Magnetic Resonance (NMR), and high-resolution mass spectrometry (HRMS). A cationic amphiphilic cyclodextrin (CD) was initially used to complex the siRNA and then co-formulated with the anionic amphiphilic CD. Characterisation of the co-formulated NPs indicated a significant reduction in charge from 34 ± 7 mV to 24 ± 6 mV (*p* < 0.05) and polydispersity index 0.46 ± 0.1 to 0.16 ± 0.04 (*p* < 0.05), compared to the cationic CD NPs. Size was similar, 161–164 nm, for both formulations. FACS and confocal microscopy, using AML cells (HL-60), indicated a similar level of cellular uptake (60% after 6 h) followed by endosomal escape. The nano co-formulation significantly reduced the charge while maintaining gene silencing (21%). Results indicate that blending of anionic and cationic amphiphilic CDs can produce bespoke NPs with optimised physicochemical properties and potential for enhanced in vivo performance in cancer treatment.

## 1. Introduction

Acute myeloid leukaemia (AML) is a malignant blood cancer with an increased incidence and poor prognosis in adults. Myeloid progenitor cells, leukaemic blasts, stop differentiating and start to proliferate uncontrollably in the bone marrow, peripheral blood, and spleen. Leukaemic myeloblasts replace functioning bone marrow [1]. Conventional therapies, which target only dividing cells, are capable of reducing the number of leukaemic cells and may be unable to eliminate the non-dividing leukaemia stem cells (LSCs). Therefore, LSCs resistant to conventional therapies, may remain in the bone marrow, and at a later stage begin to divide again causing relapse with a high mortality rate [2,3].

As an alternative approach, gene therapy targeting AML fusion transcripts has been considered promising [4]. AML has a complex genomic landscape and numerous clones are defined by or associated with mutations in key genes. Some of the best-known mutations include alterations in the *PML-RARA* and *FLT3* genes which have led to targeted therapies and improved prognosis [5]. Mutations and changes in expression are also noted in the *KMT2a* and *KAT2a* genes in AML. Pre-clinical results showed that *KAT2a* knockdown exhibited transcriptional instability in AML stem cells rather than hematopoietic stem cells, making this gene a potential target for AML cells and leukaemic stem cells directed therapy [6]. KAT2a, an epigenetic modulator is a lysine acetyltransferase overexpressed in all AML types, participating in the maintenance of leukaemia via acetylating histone H3 lysine 9. For this reason, silencing of *KAT2a* leads to deacetylation at transcription sites and consequently to impaired transcription of the gene responsible for maintaining stemness in the AML clone [7]. The knockdown interrupts self-renewal and moves the cell cycle into the differentiation state [8].

Despite early promising data, the delivery of nucleic acids into cells faces multiple challenges, including delivery across lipid bilayers and degradation of nucleic acids by endonucleases, therefore, a delivery system is required to overcome these barriers and to achieve the required duration of activity at the diseased site [9]. Additionally, blood cells have proven to be difficult to transfect compared to other cell types [10,11]. The non-viral delivery of nucleic acids utilising the nanoparticle (NP) approach is a promising tool to overcome physiological barriers [12]. A range of materials have been investigated as non-viral delivery vectors for therapeutic nucleic acids [13,14].

Previously, to overcome the obstacles in transfecting blood cells, antibody-targeted cyclodextrin (CD)-based NPs were investigated for siRNA delivery to AML LSCs. The antibody against the IL-3 receptor alpha-chain (IL-3R alpha, also known as CD123) was incorporated into the formulation for selective targeting of KG1 cells, an AML leukaemia stem and progenitor cell line. The authors demonstrated efficient delivery of bromodomain-containing protein 4 (*BRD4*) siRNA to KG1 cells and in ex vivo primary AML patient derived samples, which resulted in downregulation at mRNA and protein levels. The silencing of *BRD4* induced myeloid differentiation and triggered leukaemia apoptosis [2].

While cationic materials are required to complex negatively charged nucleic acids, a reduced cationic charge density at physiological pH is desirable to minimise in vivo toxicity [15] and improve the stability of the formulation in salt- and serum-containing media [16], an important characteristic for clinical translation. Consequently, the aim of the present study was to develop a novel delivery system with reduced charge density by co-formulating amphiphilic cationic CDs with negatively charged amphiphilic CDs. The benefits of incorporating a negatively charged entity to pre-formed cationic NPs were previously illustrated by the work of Ball et al. [17]. They assessed the co-formulation of mRNA and siRNA aiming to target diseases with concurrent abnormal gene up- and downregulation. The co-delivery of both RNAs increased gene silencing in contrast to siRNA only lipid NPs. To expand the potential of this finding to a broader range of diseases and reduce costs related to the additional RNA species, mRNA was substituted with a synthetic negatively charged polymer. Lipid NPs containing the polyanion and siRNA showed the same efficacy as the co-formulation of mRNA and siRNA. It was suggested that the inclusion of the ‘helper polymer’ could increase the stability and electrostatic interaction within the particle and also improve the efficiency of the delivery system [17]. In this study, we are aiming to simulate this positive effect with the co-formulation of an anionic amphiphilic CD.

The CDs formulated with siRNA against *KAT2a* were characterized, and the safety and efficacy of the NPs were evaluated in HL-60 cells, an AML cell line expressing the gene of interest. To our knowledge, this is the first time such a CD co-formulation approach has been investigated for siRNA delivery.

## 2. Results and Discussion

### 2.1. Synthesis of the Amphiphilic Anionic CD

The amphiphilic cationic CD was synthesised as previously described [18]. The amphiphilic anionic CD was synthesised as follows [19]: β-CD was silylated at the 6-OH position [18] in order to direct the subsequent esterification of 2-OH and 3-OH positions using dodecylanhydride. Facile de-silylation to yield compound 3 was followed by treatment with 23 equivalents sulphurtrioxide-triethylamine complex at 80 °C for 5 days in an inert atmosphere to generate the sulfate (4) as a trimethylamine salt in quantitative yield. All steps required purification by silica-based chromatography (Figure 1).

The molecular structure was confirmed by means of Fourier-transform infrared spectroscopy (FT-IR), Nuclear Magnetic Resonance (NMR), and high-resolution mass spectrometry (HRMS), with following results:

FT-IR, thin film on NaCl (cm^−1^) 3476, 2956, 2924, 2854, 1748, 1466, 1260, 1203, 1173, 1047, 1006. The spectrum peak at 1748 cm−1 is characteristic for carbonyl groups that were created during esterification of 2-OH and 3-OH positions. The peak at 1260 cm^−1^ is an indication for the successful generation of sulfate [20].

^1^H NMR, 300 MHz, CDCl_3_ ppm: 0.83 (6H, m); 1.20 (32H, m); 1.19 (9H, t, J = 7.2 Hz); 1.45 (2H, m); 2.05 (2H, m); 2.25 (4H, m); 3.15 (6H, m); 3.41 (1H, m), 3.89 (1H, t, J = 9.0 Hz); 4.15 (1H, m); 4.30 (1H, m); 4.70 (1H, dd, J = 10.2, 3 Hz); 5.15 (1H, d, J = 3 Hz); 5.25 (1H, t, J = 9 Hz); 9.15 (1H, s, D_2_O exchange) (Appendix A).

^13^C NMR, 100 MHz, CDCl_3_ ppm: 8.75, 9.71, 10.97, 14.07, 14.12, 19.69, 22.72, 23.00, 23.74, 24.65, 24.83, 27.10, 28.93, 29.41, 29.45, 29.50, 29.60, 29.68, 29.72, 29.86, 29.91, 30.36, 31.98, 34.00, 34.13, 37.11, 38.72, 46.45, 68.16, 128.81, 130.90, 132.45, 167.79, 172.52, 172.81 (Appendix A).

The molecular weight of the anionic CD C_210_H_378_O_70_S_7_ [M^+^] was calculated as 4244.4060 g/mol and confirmed to be 4244.3947 g/mol by HRMS with an error of 2.66 ppm.

### 2.2. Physicochemical Characterization

Previously cationic CDs have proven highly effective in silencing gene expression in a range of in vivo animal models of disease [21,22,23]. A reduction in the cationic charge density is likely to extend the in vivo circulation time and minimise potential aggregation and toxicity [24,25]. To help achieve this charge reduction the co-formulation of cationic with anionic amphiphilic CDs was explored. The concept of co-formulating cationic and anionic entities to enhance the performance of non-viral delivery of siRNA was reinforced by the work of Ball et al. [17]. Ball and co-workers initially co-formulated siRNA with mRNA using a blend of lipids and observed an improved gene-silencing effect compared to siRNA-only NP at the same siRNA concentration. Following this observation, mRNA was substituted with a polyanionic polymer creating lipid NPs containing only siRNA that provided the same gene silencing efficiency as the formulation with both nucleic acids. The additional negative charge helped to increase the electrostatic interaction within the particles reducing the dose of siRNA and increasing the gene silencing efficacy [17].

In this study, an anionic amphiphilic CD was synthesized and co-formulated with an amphiphilic cationic CD (Figure 2) aiming to reduce the overall positive charge, improve the stability and the gene knockdown activity.

Initially, the capacity of the amphiphilic anionic CD to neutralise the positively charged CD was assessed in the absence of siRNA (Appendix A). Neutralisation occurred between MRs of 0.85:1 to 0.9:1 (amphiphilic cationic:anionic CD), indicating that the charge density of the amphiphilic anionic CD may not be as high as that of the amphiphilic cationic CD. Therefore, 15% more anionic CD is needed to neutralise the cationic CD. 

Using this initial data, preliminary studies were performed to identify the optimum MR of both CDs with the aim of achieving NPs containing 100 nM siRNA, with a charge between −30 mV to 30 mV, size below 200 nm and PdI between 0–0.3. These physicochemical parameters are desirable in NP delivery where: negatively or positively charged particles were proven to show better in vitro uptake than neutral NPs [26]; particles below 200 nm are able to escape the reticulo endothelial system (RES) [27,28] and low PDI guarantees uniform particles [29] thus increasing the possibility to reach the site of disease. 

Optimisation studies in the presence of the negatively charged siRNA indicated a MR of 8.5:1:5.3 (cationic CD:siRNA:anionic CD) for the co-formulation (Appendix A). Higher MR of the anionic CD resulted in displacement of the siRNA (Appendix A). A summary of the physicochemical properties for both NP formulations is given in Table 1, where the formulation at MR 8.5:1 (cationic CD:siRNA) displayed surface charge of 34 ± 7 mV, size of 164 ± 44 nm, and PDI of 0.46 ± 0.1 while the co-formulation MR 8.5:1:5.3 (cationic CD:siRNA:anionic CD) exhibited a surface charge of 24 ± 6 mV, size of 161 ± 14 nm, and PDI 0.16 ± 0.04. Similar size and PDI values have been reported for lipid NPs [30]. The addition of the amphiphilic anionic CD resulted in a significant (*p* < 0.05) reduction in charge and PDI. No increase in size was detected with the co-formulation, which may indicate increased electrostatic and lipophilic interactions, thus, leading to more condensed and monodisperse particles [31].

The encapsulation efficiency of both formulations was investigated by RiboGreen^TM^ assay. The formulation showed 95% encapsulation efficiency and the co-formulation displayed approximately 75% encapsulation efficiency (Figure 3A). Comparable entrapment was reported elsewhere [32]. Binding of siRNA by both formulations was confirmed utilising gel retardation (Appendix A). Polyanions are known to displace nucleic acids in NP [33], however results indicate no displacement of the siRNA on addition of the anionic CD at the MR studied. 

In addition to the electrostatic interaction of the CDs, the potential interaction of the lipidic functionality of both CDs may also contribute to the formation and stability of the co-formulation. TEM was used to examine the size and morphology of the co-formulation and indicated the integration of both lipidic functionalities forming a bilayer-like structure (Figure 3B). Earlier publications have reported similar structural arrangement in CD-siRNA NPs [34,35]. Compared to DLS, the smaller particle size of 100 nm is likely due to the sample preparation for TEM which includes drying of the sample, leading to shrinkage of the particles.

It has been reported that materials with pKa values around 6.5 are optimal for non-viral delivery of nucleic acids, since such pKa will ensure a positive charge during formulation at acidic pH and a near neutral charge at physiological pH 7.4 [36,37]. The pKa of both CDs were estimated based on their chemical structure as 11 and 1 for the cationic and anionic CD, respectively (Figure 2). Following NP formation, the surface pKa was determined. The formulation displayed a pKa of 8.5 ± 0.1, significantly higher (*p* < 0.05) than that of the co-formulation with a pKa of 7 ± 0.35 (Figure 3C,D). Since these are surface pKa values, the reduction in the number of cationic species on the particle surface is reflected by a reduction in the surface pKa in the same way that surface charge is seen to be reduced by co-formulation.

The ability of CDs to protect the incorporated siRNA in the presence of serum was monitored. This is important considering the exposure of nucleic acids in the blood circulation and the potential for degradation [38]. Both formulations show similar behaviour in FBS over the time course studied. It is suggested that the FBS covers the surface with non-specific binding [39,40,41] which possibly led to the slight increase of 10 nm observed for both formulations immediately after FBS exposure, however, no further increase in size was detected at longer incubation times up to 24 h (Figure 4A,B), as reported previously for dendrimer nanocomplexes [32]. The protective role of the CDs was confirmed by heparin-release assay [42], in which heparin, a negatively charged polysaccharide, displaces the incorporated siRNA due to charge competition. As shown in Figure 4C, siRNA release was verified at all time points for both CD-NP formulations. For further confirmation, RNA concentrations were measured (Figure 4D). Free siRNA decreased in concentration by half in less than 3 h, while the siRNA concentration in the NP formulations remained constant, independent of the time exposed to serum. These results conclude that the siRNA released by the heparin was intact and undegraded, indicating that both NP formulations could protect the nucleic acid from enzymatic degradation in serum [32].

### 2.3. In Vitro Cell Culture

Having determined that the addition of the amphiphilic anionic CD to the formulation can significantly impact the surface charge, both formulations were evaluated using HL-60 cells. HL-60 is a human peripheral blood cell line derived from a 36-year-old woman with acute myeloid leukaemia, where the epigenetic modulator KAT2a is overexpressed [7]. The cells were incubated with the CD-based formulation carrying FAM-labeled siRNA at a final concentration of 100 nM and cellular uptake was monitored by flow cytometry and confocal microscopy.

Confocal microscopy confirmed siRNA internalization in HL-60 cells as demonstrated by the green fluorescence present around the nuclei at all imaged time points (Figure 5A). The relatively lower surface charge of the co-formulation did not alter the cellular uptake indicating that charge alone is not the sole driver for internalization [40]. In contrast, the amphiphilicity of the CDs promoting a lipophilic interaction at cellular level may also be a significant factor [43]. As shown in Figure 5B, the uptake patterns were similar for both formulations over time (6, 12 and 24 h), reaching approximately 60% after 6 h and decreasing by half after 24 h, consistent with the confocal images that show reduced siRNA signal over time. Similar behaviour has been previously published [11].

Generally, after entering the cells via endocytosis, NPs containing siRNA if trapped in early endosomes, which mature into late endosomes and ultimately fuse with lysosomes, may be degraded [44]. Therefore, effective escape from late endosomes/lysosomes is a critical step to achieve gene silencing. Herein, Lysotracker Red was used to label later endosomes/lysosomes and the intracellular trafficking was investigated by confocal microscopy. After 6 h, both formulations were located within the late endosome/lysosome, as indicated by the co-localisation of siRNA (green) and LysoTracker (red) (Figure 6, indicated by white arrows). The time frame for late endo-lysosomal release differed for the two NP formulations. After 12 h, late-endosomal/lysosomal release was seen with the formulation (Figure 6A), but not with the co-formulation which still showed co-localisation (Figure 6B). Endo-lysosomal escape for the co-formulation appeared mainly to occur at the later time of 24 h. 

The different endosomal release patterns of the two formulations may be due in part to the difference in overall surface charge and surface pK_a_. It is well known that positively charged particles more readily escape from the endosome via electrostatic interaction with the endogenous anionic components of the endosomal membrane resulting in endosomal disruption [45,46]. At endosomal pH of 5.5, the co-formulation NPs with surface pK_a_ of 7 may display less of the cationic species at the NP surface in comparison to the formulation NPs with surface pK_a_ of 8.5. This decrease may lead to weaker nanoparticle-endosomal-membrane-interaction thus explaining the slower endosomal escape of the co-formulation detected after 24 h (Figure 6).

To assess the level of gene knockdown HL-60 cells were transfected with 100 nM *KAT2a* siRNA per well for 3 days. Although blood cells are difficult to transfect with non-viral vectors [10,11], both CD-NP formulations significantly reduced mRNA levels relative to free *KAT2a* (*p* < 0.05) by 29 ± 11% for the formulation, and 21 ± 2% for the co-formulation (Figure 7), in the absence of toxicity (Appendix A). Landry and colleagues reported GFP silencing in THP-1, another leukaemia cell line, of 20–26%, indicating that the levels of gene knockdown achieved in this study are comparable [11].

The gene knockdown efficiencies for both formulations are not significantly different, however the trend for a slightly lower level seen with the co-formulation may be due to the lower loading capacity, (Figure 3A), and the slower endosomal release potentially resulting in greater endosomal degradation (Figure 6). While the co-formulation did not enhance the level of gene silencing in this in vitro study, the reduction in charge density achieved following addition of the anionic CD may be a significant advantage for reducing in vivo toxicity [15].

Work is ongoing to improve the transfection efficiency by further modifying the CD chemistry with the aim of enhancing endosomal escape. In addition, due to the flexibility of the CD basic structure potential exists to incorporate a targeting ligand to help ensure cell specific uptake in vivo. Antibody-labelled NPs have shown potential especially in the design of precision medicines for cancer therapy [47,48]. This targeting approach has also been applied to CDs, where Guo et al. [2], have shown that antibody-targeted cyclodextrin-based NPs increased the level of knockdown in KG1 cells, an AML leukemia stem and progenitor cell line, and ex vivo in patient samples by 35% relative to the untargeted formulation.

## 3. Materials and Methods

### 3.1. Materials

MISSION^®^ siRNA Universal Negative Control #1 and MISSION^®^ siRNA Fluorescent Universal Negative Control #1 6-FAM were purchased from Sigma Aldrich. Pre-designed *KAT2a* (or *GCN5*) (antisense CCUACUUGUCAAUGAGGCCtc) was obtained from ThermoFisher (Dublin, Ireland). All chemicals and materials were of analytical grade and purchased from Sigma-Aldrich (Wicklow, Ireland), unless otherwise specified.

The cationic amphiphilic CD was synthesised as previously described by using a ‘click’ chemistry approach to attach the amine group on the secondary side, the primary side was modified with lipid chains (C_12_) [18]. The anionic amphiphilic CD was synthesised as described above and the structure was confirmed by Fourier-transform infrared spectroscopy (FT-IR), Nuclear Magnetic Resonance (NMR), and high-resolution mass spectrometry (HRMS).

### 3.2. Nanoparticle Preparation

Both amphiphilic cationic and anionic CDs were dispersed in RNase-free water (0.25 mg/mL) via water bath sonication (Branson 1510) for 1 h at 70 °C, and used immediately after sonication. NPs with different mass ratios (MR) were prepared by adding the required amount of siRNA and cationic CD to RNasefree water, herein referred as the formulation. The formulation was then incubated at room temperature at 450 rpm for 30 min in a thermomixer. Following incubation, the amphiphilic anionic CD was added to the cationic CD:siRNA-complex, forming the co-formulation, which was subsequently incubated at 65 °C, 450 rpm for 30 min. The final siRNA concentration was 500 nM.

### 3.3. Physicochemical Characterisation

#### 3.3.1. Size, Poly Dispersity and Zeta Potential

Particle size and zeta potential were measured by dynamic light scattering (DLS) using the Zetasizer Nano-ZS (Malvern Instruments, Malvern, UK). Particle size (Z-Average) and poly dispersity measurements were conducted using a non-invasive backscatter system (scattered light angle of 170°) in a disposable narrow size cuvette with a volume of 500 µL. Electrophoretic mobility is used to determine the zeta potential (forward scatter angle of 12.8°). Zeta potential was measured with disposable folded capillary cells (DTS1070) with minimum required volume of 750 µL at the monomodal mode with 60 s delay between measurements. The settings were as follows: Protein as material (refractive index 1.450 and absorption 0.001), water as dispersant (viscosity 0.8872cP, refractive index 1.330) at 25 °C.

#### 3.3.2. Complexation Efficiency

The complexation efficiency was investigated using a 1% Agarose gel in Tris/Borate/EDTA buffer. Both running buffer and gel, contained SafeView (NBS Biologicals, Huntingdon, UK), a Nucleic Acid Stain (6 µL/100 mL). Free siRNA (positive control) and samples contained 3 µL BlueJuice^TM^ Gel Loading Buffer (10×) (Invitrogen, Waltham, MA, USA) and 27 µL sample. The gel was run at 90 V for 60 min (Bio-Rad, PowerPacBasic, Hercules, CA, USA) and imaged under UV with Uvitec Cambridge Mini HD.

#### 3.3.3. Encapsulation Efficiency

Encapsulation efficiency (EC) was assessed using the modified Quant-iT RiboGreen^TM^ RNA Assay kit (Invitrogen, Invitrogen, Waltham, MA, USA) [32]. Freely accessible siRNA was determined before the addition of surfactant (F_free_), encapsulated siRNA was measured upon lysis of the particles with 10% Triton X-100 (F_total_) (final Triton concentration per well: 1%). After addition of the RiboGreen reagent, Fluorescence intensity was determined using a Tecan Spark^®^ microplate reader at 485 nm excitation wavelength and 535 nm emission wavelength in a black 96-well plate (Gain 55, top reading), siRNA was quantified by means of a calibration curve. The EC was calculated using the equation:EC = [(F_total_ − F_free_)/F_total_] ∗ 100

#### 3.3.4. Determination of pK_a_

Buffers from pH 4–12 in increments of 0.5 (pH 4–6.5 citrate buffer, pH 7–8 phosphate buffer, pH 8.5–12 Tris-HCl buffer), and 2-(p-toluidino)-6-naphthalene sulfonic acid (TNS) solution at 300 µM in DMSO were prepared. In a black 96-well plate, 90 µL of buffer with 10 µL samples were mixed. Finally, 2 µL of TNS solution was added (6 µM TNS/well) and incubated at room temperature in a shaker for 5 min, protected from light. Fluorescence intensity was measured using the Tecan Spark^®^ (Männedorf, Switzerland) microplate reader at an excitation of 340 nm and an emission of 485 nm (top reading). Sigmoidal interpolation was applied to the collected data, where the inflection point (LogIC50) defined the apparent pKa of the formulations [10]. 

#### 3.3.5. Serum Stability

To assess the behaviour of the NPs in the presence of serum, samples were exposed to 10% fetal bovine serum (FBS, heat-inactivated) for different time intervals at 37 °C for up to 24 h. Particle aggregation was monitored by DLS. 

To assess the ability of the CD.NPs to prevent siRNA degradation following exposure to serum, a heparin displacement assay was used. Following incubation at 37 °C in 10% FBS, samples were treated with freshly prepared heparin (10 mg/mL, roughly 2000 I.U. in phosphate-buffered saline) for 75 min at room temperature (20% heparin in final sample). BlueJuice^TM^ Gel Loading Buffer (10×) (Invitrogen, Invitrogen, Waltham, MA, USA) was added to the samples and loaded onto the gel, containing SafeView (NBS Biologicals, Huntingdon, UK). The gel was run at 120 V for 30 min (Bio-Rad, PowerPacBasic, Hercules, CA, USA) and imaged under UV. The heparin-treated samples were measured with NanoDrop One (ThermoScientific, Dublin, Ireland) to quantify the residual siRNA.

### 3.4. Cell Culture

HL-60, a human peripheral blood cell line derived from a 36-year-old woman with acute leukaemia, was kindly donated by Cancer Research @UCC and was maintained in Iscove Modified Dulbecco Media with 20% FBS (heat-inactivated) and 1% Penicillin-Streptomycin in a humidified 37 °C incubator with 5% CO_2_. For all experiments the FBS concentration was decreased to 10%.

#### 3.4.1. Cellular Uptake

Flow cytometry was used to assess cellular uptake. Cells were seeded at 3.5 × 10^5^ cells per ml in a 24-well plate and transfected with formulations encapsulating FAM-labelled siRNA for 6, 12 and 24 h (100 nM final concentration). After incubation, cells were centrifuged at 1500 rpm for 5 min at 4 °C and the supernatant was removed. The pellet of cells was resuspended with 400 µL phosphate-buffered saline (PBS) and transferred into polystyrene round-bottom tubes (Becton Dickinson, Franklin Lakes, NJ, USA). Data were acquired on BD CELESTA Facs Analyser (minimum 40,000 events).

#### 3.4.2. Intra-Cellular Trafficking

The intracellular localization and co-localization of the formulations with lysosomes were evaluated by confocal microscopy. Cells were seeded at 5 × 10^5^ cells per ml onto Poly-D-Lysine coated coverslips in a 24-well plate and treated with FAM-labelled siRNA contatining formulations. The incubation periods varied from 6, 12 to 24 h. LysoTracker^TM^ Red DND-99 (Invitrogen, Invitrogen, Waltham, MA, USA), a red-fluorescent dye for labelling and tracking acidic organelles, was prepared at 100 nM in serum-free media. The Syto^TM^ Deep Red Nucleic Acid Stain (Invitrogen, Invitrogen, Waltham, MA, USA), which stains the nuclei of cells, was added to the Lysotracker working solution to give a final nucleic acid stain of 1 µM. After incubation, the media was removed carefully, and the dyes were added to each well (500 µL dye solution/well); after 150 min the dye solution was withdrawn. The cells were fixed with 4% paraformaldehyde for 10 min at room temperature followed by wash with PBS. The coverslips were removed and mounted with Fluorescent Mounting Media (Dako, Agilent, Santa Clara, CA, USA); the cells were imaged using an OLYMPUS FV1000 Confocal Laser Scanning Biological Microscope.

#### 3.4.3. Toxicity Assay

The cell counting kit -8 (CCK-8) assay was used to assess cell viability. In a 96-well plate, 100 µL of cell suspension (2.5 × 10^5^/mL) was transfected with 25 µL of sample (siRNA 100 nM/well), followed by incubation for 3 days at 37 °C incubator with 5% CO2. After incubation, 10 µL of CCK-8 solution was added followed by a further 4 h incubation, and absorbance was measured using a plate reader at 450 nm. Freshly prepared growth media was used as a blank.

#### 3.4.4. Gene Knockdown

Cells (3 × 10^5^ cells/mL) were transfected with siRNA against *KAT2a* (100 nM/well) in a 24-well plate for 3 days. RNA extraction was conducted using GeneElute^TM^ Total RNA Purification Kit according to manufacturer’s guideline. Complementary cDNA was generated using the High-Capacity cDNA Reverse Transcription kit (Applied Biosystems^TM^, Waltham, MA, USA). Real-time quantitative PCR (RT-qPCR) was performed using TaqMan Universal PCR Master Mix (Applied Biosystems^TM^) and BIO-RAD CFX96^TM^ Real-Time System. GAPDH was chosen as the housekeeping gene (4331182) and *KAT2a* probe with the most coverage was purchased (AM16704, TaqMan assay, Applied Biosystems^TM^). Thermal cycling conditions were as follows: 50 °C for 2 min, 95 °C for 5 min, 95 °C for 15 s, 55 °C for 30 s, 72 °C for 1 min, 40 cycles. The last cycle was finalised with a holding step for 7 min at 72 °C.

#### 3.4.5. Statistical Analysis

All statistical analysis was performed using GraphPad Prism 7. Unpaired *t*-test was used for physicochemical characterization measurements to compare the two formulations. Multiple comparison Dunnett’s post hoc test was utilized to compare all groups in cell culture experiments. Statistical significance was considered when *p* < 0.05.

## 4. Conclusions

The chemical versatility of CDs facilitates a range of structural modifications to optimize the physicochemical characteristics and subsequent efficacy of nanoparticles containing therapeutic nucleic acids. By varying the MRs of an amphiphilic cationic CD with an amphiphilic anionic CD in a co-formulation approach it is possible to modulate the size and charge of NPs containing siRNA. The co-formulation of the anionic amphiphilic CD led to a charge reduction of 10 mV and an approximate 3-fold reduction in polydispersity. While the reduction in the cationic charge is likely to be beneficial for in vivo delivery, the current in vitro studies indicate that endosomal escape maybe the main barrier to enhancing gene silencing above the levels reported (21 ± 2%). Structure-activity studies using functionalised CDs, including optimisation of the pK_a_, to enhance endosomal escape are under investigation. In summary, the data presented illustrates the potential of CDs as a delivery platform for nucleic acid therapeutics to treat cancers such as leukemia with high unmet clinical needs.

## Figures and Tables

**Figure 1 ijms-23-09791-f001:**
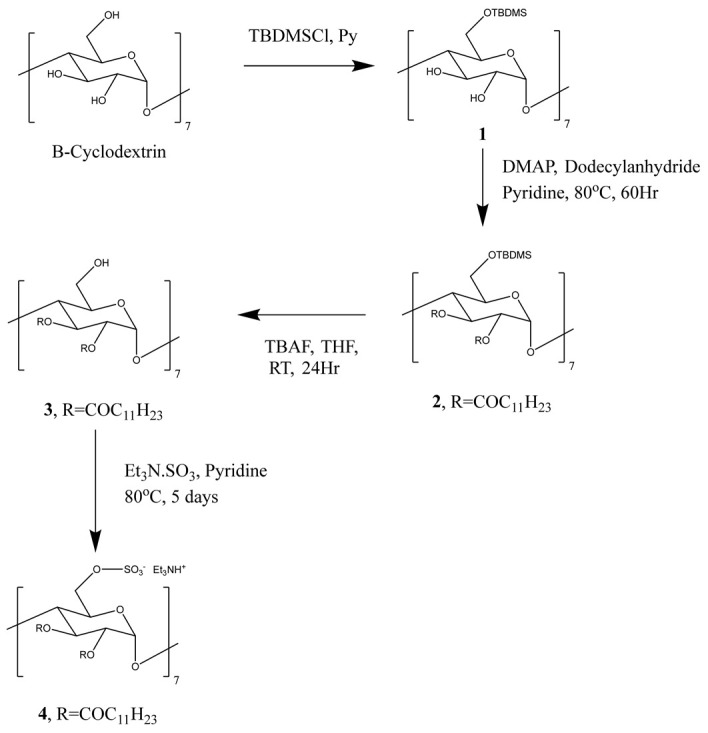
Chemical synthesis of anionic amphiphilic cyclodextrin (CD). Silylation at position 6 of the β-CD which enables the direct esterification with dodecylanhydride (C_12_) at positions 2 and 3. Subsequently, de-silylation and sulphation at position 6 produces the trimethylamine salt, resulting in a negatively charged amphiphilic CD.

**Figure 2 ijms-23-09791-f002:**
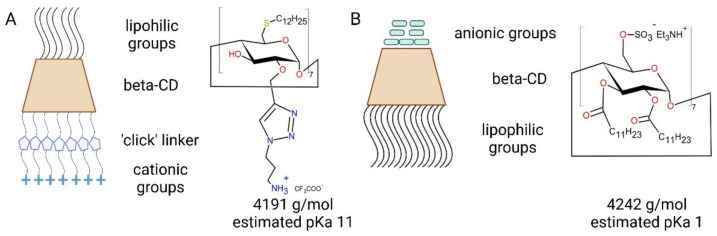
Chemical structure (mol. MW and pK_a_) and schematic of charged amphiphilic cyclodextrins (CDs). (**A**) The amphiphilic cationic CD, with lipid chains (SC_12_) on the primary side and a primary amine function on the secondary side. (**B**) The amphiphilic anionic CD, with a negative functional group on the primary side and lipid chains (SC_12_) on the secondary side.

**Figure 3 ijms-23-09791-f003:**
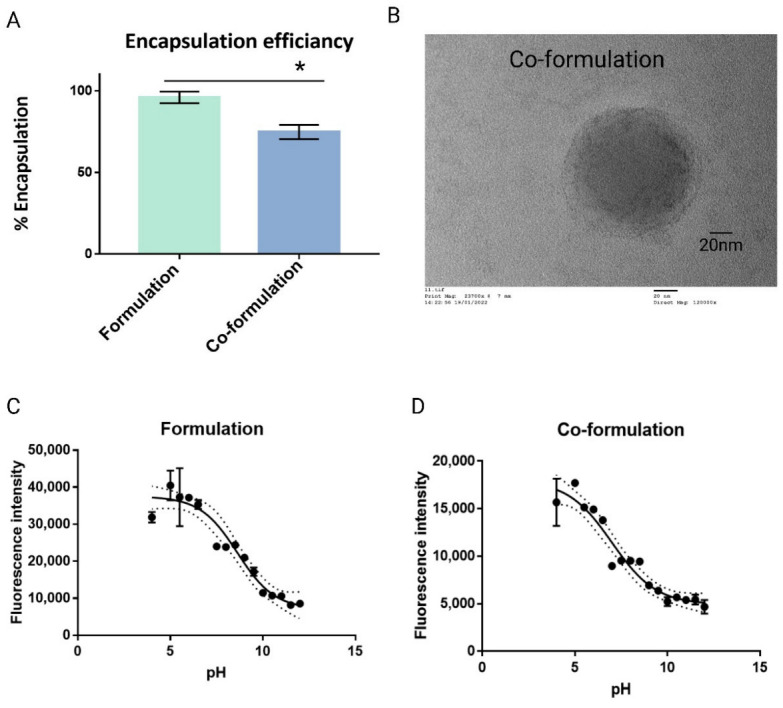
Physicochemical characterisation of cyclodextrin (CD)-based nanoparticles. (**A**) Encapsulation efficiency using Quant-iT RiboGreen^TM^ RNA Assay. (**B**) Transmission Electron Microscopy (TEM) of the co-formulation with MR 8.5:1:5.3 (cationic CD:siRNA:anionic CD). Magnification 120,000×; * *p* < 0.05, unpaired *t*-test. (**C**,**D**) Acid dissociation constant (pKa) determination of the NPs by means of 2-(p-toluidino) naphthalene-6-sulfonic acid (TNS) assay.

**Figure 4 ijms-23-09791-f004:**
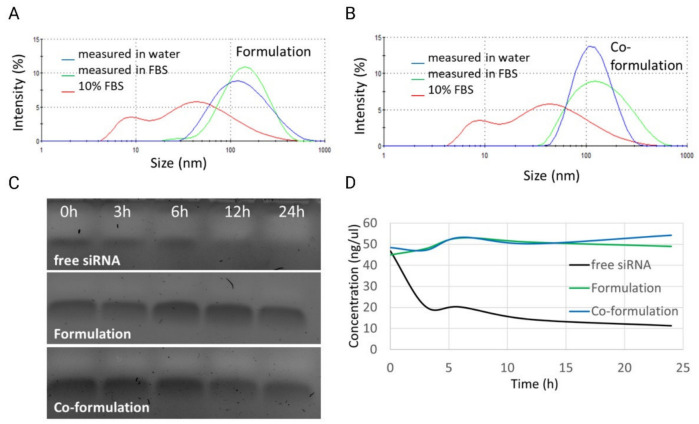
CD-NP stability in serum. (**A**) Particle size measurement after 24 h incubation in 10% serum for formulation MR 8.5:1 (cationic CD:siRNA). (**B**) Particle size measurement after 24 h incubation in 10% serum for formulation co-formulation MR 8.5:1:5.3 (cationic CD:siRNA:anionic CD). (**C**) siRNA displacement assay of serum-treated NPs with polyanion Heparin to confirm the ability to protect siRNA in CD-NP. (**D**) Quantification of siRNA of serum- and heparin-treated NP with NanoDrop One.

**Figure 5 ijms-23-09791-f005:**
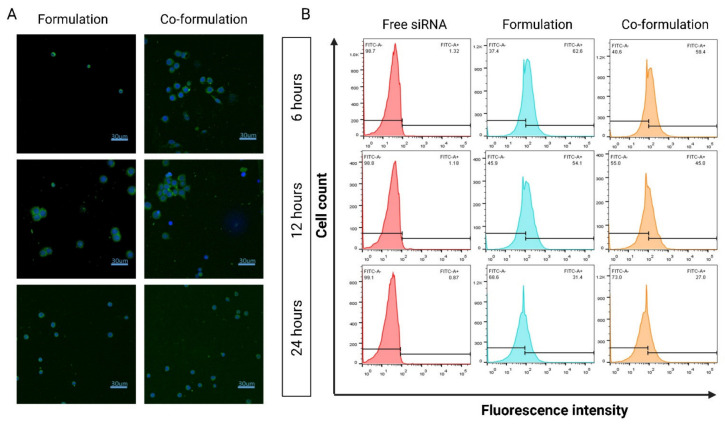
Cellular Uptake. (**A**) Confocal images of uptake into HL-60 cells with NPs incorporating FAM-labelled siRNA (green). Magnification 40×. (**B**) Flow cytometric analysis of HL-60 cells treated with different CD-based NP formulations at 6, 12 and 24 h. Formulation MR 8.5:1 (cationic CD:siRNA) and co-formulation MR 8.5:1:5.3 (cationic CD:siRNA:anionic CD).

**Figure 6 ijms-23-09791-f006:**
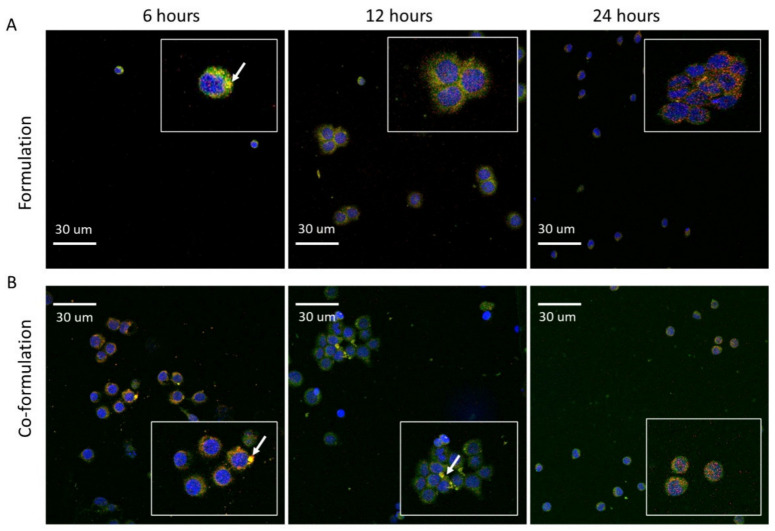
Confocal microscopy showing *KAT2a* siRNA intracellular trafficking. HL-60 cells were treated for 6, 12 and 24 h with CD-based NP formulations containing *KAT2a* siRNA (100 nM siRNA concentration per well). The fluorescent markers used included: FAM-labelled siRNA (green) positive cells with the lysosomes stained by LysoTracker^TM^ Red DND-99 (red), and the nuclei (blue) by Syto^TM^ Deep Red Nucleic Acid Stain). (**A**) Formulation, (**B**) Co-formulation. Magnification 40×.

**Figure 7 ijms-23-09791-f007:**
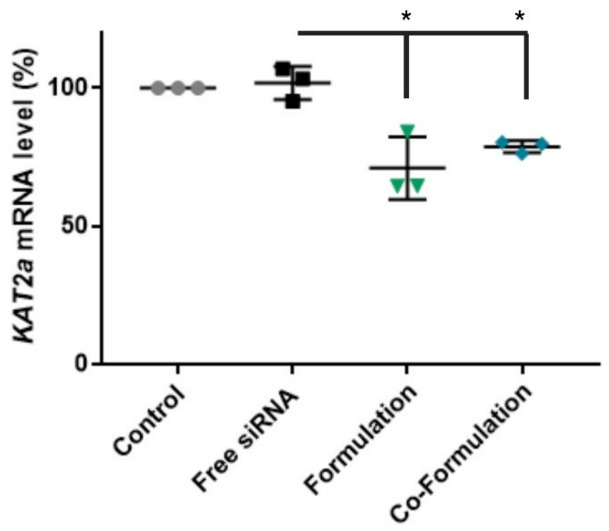
*KAT2a* mRNA levels after gene knockdown. mRNA levels of *KAT2a* (as percentage relative to the control (grey dots)) in HL-60 cells measured 72 h post-transfection by TaqMan qPCR and normalized to untreated cells. Mean ± S.E.M of 3 technical replicates performed in triplicate (siRNA: black squares; Formulation: green triangles, Co-formulation: blue diamond). One-way ANOVA with Dunnett’s post hoc test was used. * *p* < 0.05 compared to untreated group.

**Table 1 ijms-23-09791-t001:** Summary of physicochemical characterisation of CD:siRNA NPs.

	Size (nm)	PDI	Charge (mV)
Formulation (cationic CD:siRNA)	164 ± 44	0.46 ± 0.1	34 ± 7
Co-formulation (cationic CD:siRNA:anionic CD)	161 ± 14	0.16 ± 0.04	26 ± 4

## Data Availability

Not applicable.

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
