# Peer review of "Co-Formulation of Amphiphilic Cationic and Anionic Cyclodextrins Forming Nanoparticles for siRNA Delivery in the Treatment of Acute Myeloid Leukaemia"

_ijms, 2022, doi:10.3390/ijms23179791_

Round 1

Reviewer 1 Report

The authors proposed a strategy aiming at improving the delivery of siRNA through adding anionic cyclodextrins after formation siRNA/cationic cyclodextrins complex. But after reading the whole manuscript, I can not find any evidence to support the advantage of adding anionic cyclodextrins.

1. Size and protein binding profiles are almost same after adding anionic cyclodextrins. Endosomal escape and gene knockdown actually decreased. So what kind of advantage of adding anionic cyclodextrins? 

2. The authors should prove advantageous of adding anionic cyclodextrins by giving some results/evidence. This is key point of this study. How is hemolysis? 

3. In the introduction, based on research background, the authors should discuss more on how to advance the functionality of drug delivery by using the proposed strategy. Some references might be informative (e.g., https://doi.org/10.1021/jacs.0c09029).

4. The authors claimed that "No increase in size was detected with the co-formulation, which may indicate increased electrostatic and lipophilic interactions, thus, leading to more condensed and monodisperse particles. This was confirmed by the significant decrease in PDI (p < 0.05).". Why more condensed particle will lead to significant decrease in PDI? No logic relationship between compaction and PDI. The possible reason is that the anionic cyclodextrins dissociate the complex, leading to re-self-assembly. In addition, FRET is suggested to apply to evaluate the more compact structure.

Author Response

We thank you the reviewer for the comments and suggestions, we have replied to each point below.

Comments and Suggestions for Authors

The authors proposed a strategy aiming at improving the delivery of siRNA through adding anionic cyclodextrins after formation siRNA/cationic cyclodextrins complex. But after reading the whole manuscript, I cannot find any evidence to support the advantage of adding anionic cyclodextrins.

  1. Size and protein binding profiles are almost same after adding anionic cyclodextrins. Endosomal escape and gene knockdown actually decreased. So, what kind of advantage of adding anionic cyclodextrins?

R: One of the main goals of adding the anionic CD was to decrease the overall cationic charge of the siRNA.cationic CD complex. Such a reduction in cationic charge is essential to help decrease the toxicity of these NPs in vivo. Co-formulation with the anionic amphiphilic CD successfully achieved this aim displaying a statistically significant charge reduction of 10 mV and an approximate 3-fold reduction in polydispersity.

  1. The authors should prove advantageous of adding anionic cyclodextrins by giving some results/evidence. This is key point of this study. How is haemolysis?

R: Please see reply above under question 1. It is well known in the literature that cationic NPs can interact with certain plasma proteins thus causing potential side effects in vivo. This current work is a pre-clinical in vitro study which aims to show that by specific formulation approaches, e.g., the co-formulation using the anionic and cationic CDs, the overall charge of the NP can be modulated. Future work to include in vivo studies is planned.

  1. In the introduction, based on research background, the authors should discuss more on how to advance the functionality of drug delivery by using the proposed strategy. Some references might be informative (e.g., https://doi.org/10.1021/jacs.0c09029)

R: This ref has now been added to the introduction as follows (line 60-61):

A range of materials have been investigated as non-viral delivery vectors for therapeutic nucleic acids [13, 14].

  1. The authors claimed that "No increase in size was detected with the co-formulation, which may indicate increased electrostatic and lipophilic interactions, thus, leading to more condensed and monodisperse particles. This was confirmed by the significant decrease in PDI (p < 0.05).". Why more condensed particle will lead to significant decrease in PDI? No logic relationship between compaction and PDI. The possible reason is that the anionic cyclodextrins dissociate the complex, leading to re-self-assembly. In addition, FRET is suggested to apply to evaluate the more compact structure.

R: We thank the reviewer for this comment and agree that such a relationship is not valid consequently, we have now deleted the statement ‘This was confirmed by the significant decrease in PDI (p < 0.05) (line 179-180). ‘and PDI’ was added in line 177 instead.

However, we have studied the potential for siRNA displacement following the addition of the anionic CD and our results show that siRNA is displaced only at higher MR and not at the lower MRs used in the current study. This has been explained in the manuscript as follows (lines 168-170):

‘Optimisation studies in the presence of the negatively charged siRNA indicated a MR of 8.5:1:5.3 (cationic CD:siRNA:anionic CD) for the co-formulation (Supplementary data, Fig. S4). Higher MR of the anionic CD resulted in displacement of the siRNA (Supplementary data, Fig. S3)’.

Reviewer 2 Report

This work is interesting for researchers. However, the manuscript has minor comments:

1. Rewrite the abstract in the order of (a) the overall purpose of the study(b) the basic design of the study(c)major findings.

2. There is lack of research gap.

3. Make clear objectives.

4. The results should be compared with the recent literature in results and discussion section of the manuscript. 

4."The cationic amphiphilic CD was synthesized as previously described [16]." on line number 306 of page 10 needs to make more explanation. 

5. Rewrite the conclusion by including exact findings from the results and discussion. 

Author Response

We thank you the reviewer for the comments and suggestions, we have replied to each point below.

Comments and Suggestions for Authors

This work is interesting for researchers. However, the manuscript has minor comments:

  1. Rewrite the abstract in the order of (a) the overall purpose of the study, (b) the basic design of the study and (c) major findings.

R: We have rewritten the abstract as follows (Page 1):

Line 17-21: Characterisation of the co-formulated NPs indicated a significant reduction in charge from 34 ± 7 mV to 24 ± 6 mV (p < 0.05) and polydispersity index 0.46 ±0.1 to 0.16 ± 0.04 (p < 0.05), compared to the cationic CD NPs. Size was similar, 161 – 164 nm, for both formulations. FACS and confocal microscopy, using AML cells (HL-60), indicated a similar level of cellular uptake (60% after 6 hours) followed by endosomal escape.

Line 22-23 was removed.

Line 24: The nano co-formulation significantly reduced the charge while maintaining gene silencing (21%).

  1. There is lack of research gap.

R: The novelty of the current work has been clarified (line 91-92) by insertion of the following statement: 'To our knowledge this is the first time such a CD co-formulation approach has been investigated for siRNA delivery.'

3. Make clear objectives.

R: The objective of the work is clarified in the introduction as follows (line 75-77):

‘Consequently, the aim of the present study was to develop a novel delivery system with reduced charge density by co-formulating amphiphilic cationic CDs with negatively charged amphiphilic CDs’.

4.The results should be compared with the recent literature in results and discussion section of the manuscript.

R: We added various references in the result and discussion section, as follow:

Page 5, line 175-176: Similar size and PDI values have been reported for lipid NPs [30]. 

Page 5, line 186-187: Comparable entrapment was reported elsewhere [32].

Page 6, line 195-196: Earlier publications have reported similar structural arrangement in CD-siRNA NPs [34, 35].

Page 7, line 222-223: […] however, no further increase in size was detected at longer incubation times up to 24 h (Figure 4 A, B), as reported previously for dendrimer nanocomplexes [32].

Page 7, line 232: reference [32] was added

Page 9, line 298-302: HL-60 cells are in the early stage of differentiation and therefore, transfection is more difficult compared to for example THP-1 cells (a more differentiated AML cell line). Landry and colleagues reported GFP silencing in THP-1 of 20-26%, indicating that the levels of gene knockdown achieved in this study are comparably stronger [11].

  1. “The cationic amphiphilic CD was synthesized as previously described [16]." in line number 306 of page 10 needs to make more explanation.

R: Following information was added on page 10, line 331-333:

‘The cationic amphiphilic CD was synthesised, as previously described, by using a ‘click’ chemistry approach to attach the amine group on the secondary side, the primary side was modified with lipid chains (C12).

  1. Rewrite the conclusion by including exact findings from the results and discussion.

R: The conclusions have been modified by the addition of the following:

Line 468-470: The co-formulation of the anionic amphiphilic CD led to a charge reduction of 10 mV and an approximate 3-fold reduction in polydispersity.

Line 472: above the levels reported (21 ±2%).

Reviewer 3 Report

The author, Ayse Kont has reported Co-formulation of amphiphilic cationic and anionic cyclodex-

trins forming nanoparticles for siRNA delivery in the treatment of acute myeloid leukaemia. The manuscript has well writeen and idea is good but there many concern before acceptnace.

1. Figure 3. Transmission 181 Electron Microscopy (TEM) of the co-formulation with MR 8.5:1:5.3 (cationic 182 CD:siRNA:anionic CD). why author has not take TEM without loading?

2. Which ratio is best for successful delivery ? did author done orthogonal test?

3. There must be random combination as orthogonal test plz show in SI for reader

4. For these test cite paper. DOI: 10.1166/jnn.2016.11908,  DOI: 10.1007/s11051-014-2347-9.

5. Have author test in both vitro and invivo? for concern see ,invitro, DOI: 10.1166/jnn.2014.8201

in vivo.  DOI: 10.1021/acsomega.9b00119

6. Author has test any enzymes for double checking physiological parameter?

7. Where are profile results of Nuclear Magnetic Resonance (NMR),???

Author Response

We thank you the reviewer for the comments and suggestions, we have replied to each point below.

Comments and Suggestions for Authors

The author, Ayse Kont has reported Co-formulation of amphiphilic cationic and anionic cyclodextrins forming nanoparticles for siRNA delivery in the treatment of acute myeloid leukaemia. The manuscript has well written, and idea is good but there many concerns before acceptance.

1.Figure 3. Transmission Electron Microscopy (TEM) of the co-formulation with MR 8.5:1:5.3 (cationic CD:siRNA:anionic CD). Why has author not taken TEM without loading?

R: The electrostatic interaction between the negatively charged nucleic acid, siRNA, and the cationic CD is essential for the formation of the NP hence TEMs of the unloaded formulation are not possible.

  1. Which ratio is best for successful delivery? Did author do orthogonal test?

R: A Design of Experiment with Minitab was used to provide initial estimates of the NP mass ratios. The following explanation and graphic were added to Figure S3 in SI.

A 3-factor, 3-level Box Behnken design [49, 50] (Minitab 17) was utilised to optimize the MR of the siRNA and both CDs. Physicochemical characteristics including size, polydispersity index (PDI) and surface charge (dependent variables), were measured using dynamic light scattering (DLS) to investigate how independent variables (siRNA, amphiphilic cationic CD, and amphiphilic anionic CD quantity) affect the formulation development. Contour plots were used to indicate the optimal MR for the NPs for the in vitro cell culture studies. In vitro tests were conducted with the two formulations before and after adding the anionic CD.

  1. There must be random combination as orthogonal test plz show in SI for reader

R: Please see response above under Question 2. Figure was added to supplementary information.

Figure S3. DoE assisted optimisation of mass ratio using Minitab Statistical Software. A 3-factor, 3-level Box Behnken design [49] (Minitab 17) was used. A contour plot with a hold value for siRNA at 5.2 µg (aiming 100 nM siRNA/well) and boundaries for charge between -30 mV to 30 mV, size up to 200 nm and PdI between 0-0.3 were used to determine the optimum mass ratio.

  1. For these tests cite paper. DOI: 10.1166/jnn.2016.11908, DOI: 10.1007/s11051-014-2347-9.

R: DOI: 10.1007/s11051-014-2347-9 is now cited in supplementary information Figure S3 (Question 2).

  1. Have author test in both vitro and in vivo? for concern see, in vitro, DOI: 10.1166/jnn.2014.8201, in vivo. DOI: 10.1021/acsomega.9b00119

R: This study included only in vitro data. Future work to include in vivo studies is planned.

  1. Author has tested any enzymes for double checking physiological parameter?

R: As stated above, point 4, this study included only in vitro data. Future work to include in vivo studies and physiological parameters is planned.

  1. Where are profile results of Nuclear Magnetic Resonance (NMR)?

The NRM profiles were added to the supplementary data Fig. S1.

Round 2

Reviewer 1 Report

Although the authors did not provide the key evidence to support the advantage of adding anionic CD, reasonable explanation is there. I recommended its publication. 

Reviewer 3 Report

Revised Well